# Assessing Uncertainties of Life-Cycle CO$_2$ Emissions Using Hydrogen Energy for Power Generation

**Akito Ozawa \*** and **Yuki Kudoh**

Global Zero Emission Research Center, National Institute of Advanced Industrial Science and Technology, 16-1 Onogawa, Tsukuba 305-8569, Japan; kudoh.yuki@aist.go.jp
**\*** Correspondence: akito.ozawa@aist.go.jp; Tel.: +81-29-861-8305

**Abstract:** Hydrogen and its energy carriers, such as liquid hydrogen (LH$_2$), methylcyclohexane (MCH), and ammonia (NH$_3$), are essential components of low-carbon energy systems. To utilize hydrogen energy, the complete environmental merits of its supply chain should be evaluated. To understand the expected environmental benefit under the uncertainty of hydrogen technology development, we conducted life-cycle inventory analysis and calculated CO$_2$ emissions and their uncertainties attributed to the entire supply chain of hydrogen and NH$_3$ power generation (co-firing and mono-firing) in Japan. Hydrogen was assumed to be produced from overseas renewable energy sources with LH$_2$/MCH as the carrier, and NH$_3$ from natural gas or renewable energy sources. The Japanese life-cycle inventory database was used to calculate emissions. Monte Carlo simulations were performed to evaluate emission uncertainty and mitigation factors using hydrogen energy. For LH$_2$, CO$_2$ emission uncertainty during hydrogen liquefaction can be reduced by using low-carbon fuel. For MCH, CO$_2$ emissions were not significantly affected by power consumption of overseas processes; however, it can be reduced by implementing low-carbon fuel and waste-heat utilization during MCH dehydrogenation. Low-carbon NH$_3$ production processes significantly affected power generation, whereas carbon capture and storage during NH$_3$ production showed the greatest reduction in CO$_2$ emission. In conclusion, reducing CO$_2$ emissions during the production of hydrogen and NH$_3$ is key to realize low-carbon hydrogen energy systems.

**Keywords:** hydrogen energy; power generation; supply chain; life-cycle inventory analysis; Monte Carlo simulations

## 1. Introduction

Hydrogen energy (hydrogen and its energy carriers such as liquid hydrogen (LH$_2$), methylcyclohexane (MCH), and ammonia (NH$_3$)) are attracting attention for realizing carbon neutrality and preventing global warming. Hydrogen does not emit carbon dioxide (CO$_2$) during the utilization phase and can be produced with low-carbon emissions through water electrolysis, or by combining natural gas reforming and coal gasification processes with carbon capture and storage (CCS) technology. Therefore, energy-related CO$_2$ emissions can be drastically reduced by replacing fossil fuel combustion with low-carbon hydrogen utilization. According to the International Energy Agency's roadmap to reach net-zero global emissions by 2050 [1], global hydrogen and hydrogen-based fuel use will expand from 87 Mt in 2020 to 528 Mt in 2050, and 98% of hydrogen in 2050 will be produced from low-carbon methods.

Innovation is essential for establishing a hydrogen economy. Japan is one of the leading countries in research, development, and demonstration (RD&D) of hydrogen technologies. In December 2017, the Japanese government formulated the Basic Hydrogen Strategy [2] which provides a shared vision for the public and private sectors to realize a hydrogen society by 2050. Considering this approach, the Strategic Road Map for Hydrogen and Fuel Cells was revised in March 2019. This scheme presented future milestones and action plans to achieve the new targets indicated in the Basic Hydrogen

Strategy. For both the strategy and road map, Japan set the future price of hydrogen at 30 JPY/Nm$^3$ by 2030 and 20 JPY/Nm$^3$ by 2050. To achieve the target price, hydrogen would need to be imported from overseas renewable electricity or be produced from cheap, unused energy resources with CCS. Japan plans to develop commercial-scale hydrogen supply chains by 2030 to procure 300 kt of hydrogen annually. Liquid hydrogen energy carriers (e.g., LH$_2$, MCH, and NH$_3$) are suitable for international transportation due to their advantage of easy handling and high energy storage compared to gaseous hydrogen. Demonstration projects for the hydrogen supply chains using liquid hydrogen energy carriers is underway [3–5]. On the other hand, the use of hydrogen energy in the electricity sector has attracted attention from the perspective of its end use [6–8]. Replacing existing thermal power generation with hydrogen energy generation is expected to reduce carbon emissions as well as serve as a regulated power supply and backup power source required for expanding renewable energy [2]. As hydrogen energy generation consumes a large quantity of hydrogen constantly, it is positioned as a technology that will be commercialized by 2030, in parallel with the construction of an international supply chain. A demonstration experiment conducted in Kobe in April 2018 achieved the world's first cogeneration of heat and electricity using a gas turbine that exclusively burns hydrogen [9]. Among the energy carriers, NH$_3$ can be directly combusted to generate electricity. NH$_3$ co-firing experiments in coal-fired power plants [10] and the development of low nitrogen oxide (NO$_x$) NH$_3$ gas turbines [11] are in progress.

Evaluation of hydrogen technologies is also important for establishing a low-carbon hydrogen society. The world's first ministerial-level meeting on realizing a hydrogen society, termed the Hydrogen Energy Ministerial Meeting, was held in Tokyo on 23 October 2018. The Chair's summary of the meeting, known as the Tokyo Statement [12] stresses the importance of investigating and evaluating the potential of hydrogen to reduce CO$_2$ and other pollutants in realizing a hydrogen society. As mentioned above, hydrogen does not emit CO$_2$ during the utilization phase; however, CO$_2$ emissions are generated by the resource and utility inputs (electricity, fuel, water, etc.) during the production, storage, and transportation stages of hydrogen and energy carriers. Therefore, to demonstrate the environmental superiority of hydrogen energy, it is necessary to evaluate the emissions of the entire supply chain through a life-cycle inventory (LCI) analysis. LCI analysis for hydrogen supply chains has been conducted in many studies on the well-to-tank greenhouse gas (GHG)/CO$_2$ emissions of hydrogen for mobility use [13–19]. In terms of LCI analysis of other applications, Ozawa et al. [20] analyzed life-cycle CO$_2$ emissions of hydrogen and NH$_3$ power generation, and Chisalita et al. [21] conducted a cradle-to-gate environmental assessment of European NH$_3$ production considering various hydrogen supply chains.

Although these studies could provide some implications for establishing low-carbon hydrogen supply chains, they rarely deal with the uncertainties of parameters related to supply chains. Since most technologies that compose hydrogen supply chains are in the development and demonstration stages, there may be a large degree of uncertainty about the environmental value of the entire supply chain at the practical application stage. Such analysis can provide researchers and policymakers with useful feedback that can help in robust decision-making regarding RD&D of hydrogen technologies. LCI analysis often involves assessing the probable impact of several variables in the target supply chains on life-cycle GHG/CO$_2$ emissions through Monte Carlo simulations [22]. However, LCI analysis that considers the uncertainties in life-cycle GHG/CO$_2$ emissions from hydrogen supply chains is limited to studies that focus on the use of hydrogen for smobility [13,19].

In most LCI studies, specific values are assumed for supply chain parameters [14–18,20,21]. However, essential technologies that compose hydrogen supply chains for power generation are still in the development and demonstration stages, and we recognize that their specification (e.g., energy input and CO$_2$ emissions intensity) at the practical application stage should have a certain degree of uncertainty, which have not been fully considered in previous research. This study aims to evaluate uncertainties regarding the environmental

benefits of hydrogen in the electricity sector by applying the Monte Carlo method for LCI analysis. While hydrogen energy is attracting attention in the context of lowering the carbon emissions of energy systems in Japan, its environmental effects can range depending on the future performance of elements that make up hydrogen supply chains. The purpose of this study is to identify and understand the important factors involved in electricity generation using hydrogen energy by conducting a stochastic LCI analysis using Monte Carlo simulations. The variations in the parameters related to the operation of hydrogen supply chains were collected by literature surveys, and the uncertainties involved in life-cycle $CO_2$ emissions from hydrogen and $NH_3$ power generation were estimated. Please note here that the life-cycle $CO_2$ emissions attributed to capital goods necessary to operate the hydrogen supply chain were out of the scope of this study. The results obtained are compared with those of coal- and liquefied natural gas (LNG)-fired power generation, and the effect of the use of hydrogen energy on reducing $CO_2$ emissions over the entire life-cycle is discussed.

## 2. Method and Assumptions

### 2.1. Outline

Figure 1 illustrates the research methodology of this study. Firstly, the system boundaries are defined to determine which unit processes to be included in this study. Secondly, inventory data related to the operation of the hydrogen supply chains are collected by literature survey. This study focuses on the life-cycle $CO_2$ emissions due to the operation of the supply chains and the emissions for capital goods are out of scope. Thirdly, based on the collected data, the probability distributions of energy input and $CO_2$ emission intensity are set for each process in the supply chains. Fourthly, according to the given probability distributions, Monte Carlo simulations are performed to calculate the life-cycle $CO_2$ emissions of hydrogen and $NH_3$ power generation stochastically. Finally, the calculated life-cycle $CO_2$ emissions are compared with those of thermal power generation to discuss the environmental benefits of hydrogen energy for power generation.

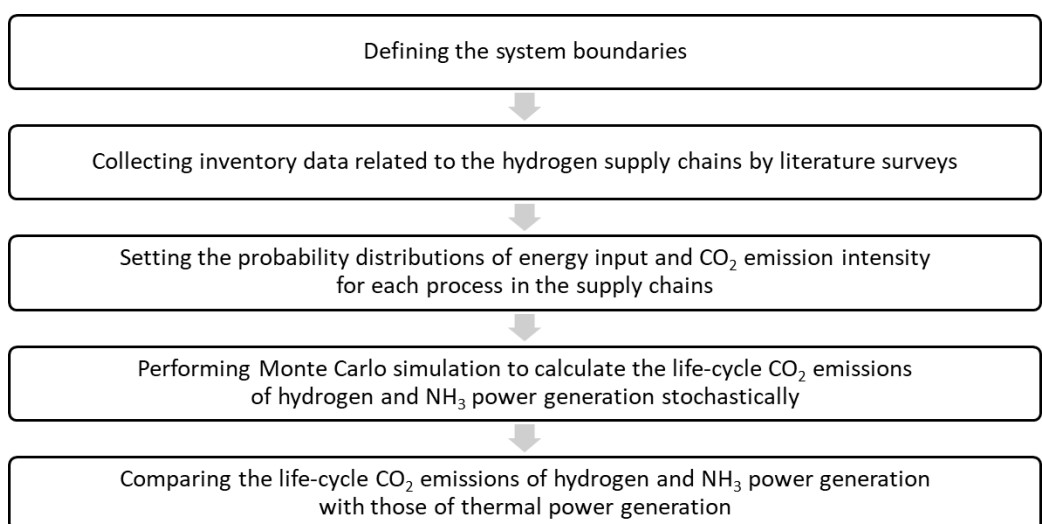

**Figure 1.** Process flow charts showing stages within the system's boundaries.

Figure 2 shows the power generation technologies and energy supply chain envisioned in this study. Respectively, both hydrogen and $NH_3$ power generation (co-firing and mono-firing) are the subjects of analyses, whereas coal- and LNG-fired power generation are used for comparison. The hydrogen supply chain sequentially consists of hydrogen production from renewable energy sources overseas; production of energy carriers; storage at the loading port ($LH_2$ or MCH); international transportation; storage at the unloading port; and finally, hydrogen restoration. In contrast, the $NH_3$ supply chain consists of its production from renewable energy or natural gas overseas; storage at the loading port;

international transportation; storage and transportation at the unloading port; and re-evaporation. The supply chain for coal and LNG consists of mining and pre-processing; domestic transportation; and international transportation. The $CO_2$ emission reduction effect of applying CCS technology was also evaluated for $NH_3$ production from natural gas and coal- and LNG-fired power generation.

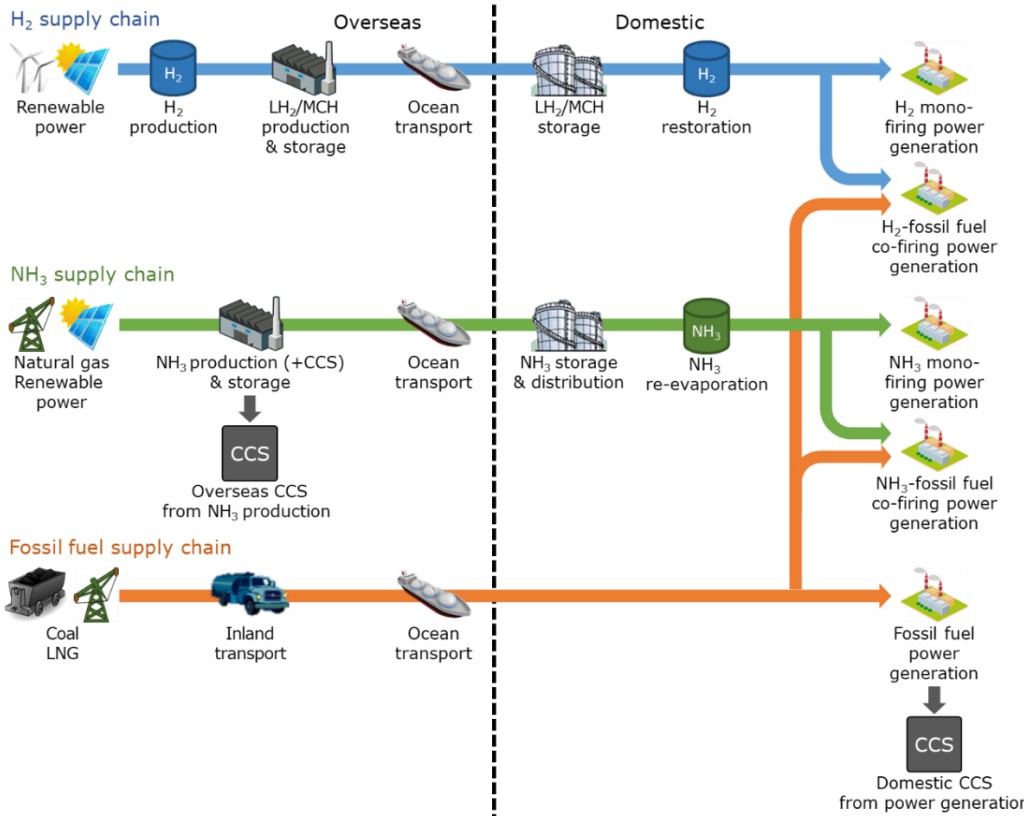

**Figure 2.** Process flow charts showing stages within the system's boundaries.

Regarding cost analyses of hydrogen and energy carriers [23,24], Australia (AUS), Norway (NOR), and the United Arab Emirates (UAE) were selected as the hydrogen-producing countries; and the UAE as the $NH_3$-producing country. The one-way distances between NOR, AUS, the UAE, and Japan were considered as 20,000 km; 10,000 km; and 12,000 km, respectively.

The LCI analysis conducted in this study considers the entire life-cycle of products and services (the series of processes from resource extraction to manufacturing, distribution, use, and disposal), and quantifies the consumption of resources and utilities, as well as the emission of environmentally hazardous substances in each process. Life-cycle $CO_2$ emissions, $E$, which is the result of the analysis, can be obtained using Equation (1):

$$E = \sum_{i,j} \left( c_{i,j} \times e_i \right) \tag{1}$$

where $c_{i,j}$ is the consumption of input $i$ in process $j$, and $e_i$ is the $CO_2$ emission intensity due to the consumption of $i$. In this study, the probability distribution of each parameter was set according to the procedure provided in Appendix A to perform an uncertainty analysis of $CO_2$ emissions from overseas renewable energy generation and by using the energy consumption of each process as variable factors. The Japanese LCI Inventory Database for Environmental Analysis (IDEA) v2.2 [25]—the largest life-cycle inventory database in Japan which has been developed by National Institute of Advanced Industrial Science and Technology (AIST), Tsukuba, Japan—was used for domestic processes to estimate the $CO_2$ emission intensity of consumption by other utilities. It has been developed

since 2008 by the Safety Science Research Division of the National Institute of Advanced Industrial Science and Technology and is packaged with data of ~3800 processes based on the Japanese Standard Industrial Classification. To estimate the $CO_2$ emission intensity of overseas processes, the 2015 statistics of the International Energy Agency [26] were used for electricity inputs and direct $CO_2$ emission factors were used as fuel inputs, while IDEA v2.2 values were used for non-energy inputs.

### 2.2. Overseas Renewable Energy Power Generation

In this study, onshore wind (NOR and AUS) and solar photovoltaics (PVs) (AUS and the UAE) were assumed to be renewable energy resources. Life-cycle $CO_2$ emissions per kWh of electricity at the transmission end were calculated on the assumption that the basic components for constructing a power plant are procured in the hydrogen- and $NH_3$-producing countries, whereas other components (nacelle and hub of wind turbine, solar panels, etc.) are manufactured in Japan and exported to hydrogen- and $NH_3$-producing countries. The probability distribution for this is listed in Appendix A Table A1. Inputs to the power plant ($x_{i,j}$) were based on data from Ozawa et al. [20]. The facility utilization rate, in-service rate, and service life of the power plant were set assuming overseas wind and sunshine conditions [20,23,27]. The specifications of the power plant are listed in Table 1. Parameters listed in Table 1 were set by collecting data from literature surveys [20].

**Table 1.** Specifications of renewable power plants.

| Parameter [Unit] | Wind | Solar PV |
|:---:|:---:|:---:|
| Capacity [MW] | 40 | 10 |
| Capacity factor [%] | 35 | 20 |
| Auxiliary power ratio [%] | 10 | 3 |
| Lifetime [year] | 30 | 30 |

### 2.3. Hydrogen Supply Chain

For the "base case" (denoted as Base in the figures) of this study, it was assumed that only the hydrogen production process uses renewable energy power generation, while the electricity input to the other overseas processes uses grid electricity. However, for countries such as AUS and the UAE, where the $CO_2$ emission intensity of grid electricity is large, the emission from the entire supply chain can be decreased by substituting overseas processes with available renewable energy electricity into the grid electricity input. Therefore, in the analysis, it was also considered that for the hydrogen supply chain from AUS and the UAE, the electricity necessary to operate the overseas processes is from the same renewable energy sources as renewable hydrogen production—denoted the "low-carbon case" (LC) in the figures. It was also assumed that the electricity input into domestic processes was supplied by hydrogen power generation.

#### 2.3.1. Hydrogen Production from Renewable Energy

It is expected that hydrogen produced from renewable energy should play a major role in the future hydrogen economy. In 2050, more than 300 Mt of green hydrogen will be produced by water electrolysis to achieve global carbon-neutrality by 2050 [1]. In this study, hydrogen is produced from onshore wind or solar PV power via alkaline or solid polymer water electrolysis. The probability distribution of electricity consumption was set as shown in Appendix A Table A2 based on published data [28,29], and the consumption of pure water was estimated to be 0.804 kg/$Nm^3$-$H_2$ based on the theoretical value.

#### 2.3.2. $LH_2$ Supply Chain

Figure 3 shows the supply chain for the use of $LH_2$ as an energy carrier. Hydrogen gas is liquefied to produce $LH_2$, which is stored in insulated tanks at the loading site, and then transported from the hydrogen-producing country to Japan by an $LH_2$ tanker (capacity: 160,000 $m^3$ at cruising speed: 16 knots). Because detailed information on the

energy consumption performance of the $LH_2$ tanker during navigation was not available, its $CO_2$ emission intensity was estimated with reference to LNG tanker transport in IDEA v2.2, it was assumed that heavy fuel oil (HFO) and $LH_2$ boil-off gas were used as marine fuel. Here, $CO_2$ emissions associated with the transportation of $LH_2$ by tankers consider the unladen voyage of the tanker from Japan to the hydrogen-producing country and its return from the hydrogen-producing country to Japan laden with $LH_2$. The imported $LH_2$ is stored in insulated tanks at the unloading port and supplied to the power plant. The probability distributions of electricity consumption due to hydrogen liquefaction and $LH_2$ storage, and the $LH_2$ boil-off rate due to international transportation were set based on published data as shown in Appendix A Table A2 [18,30–36].

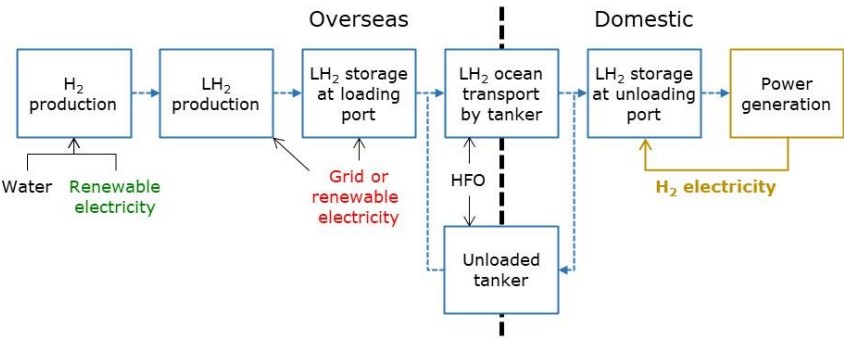

**Figure 3.** $H_2$ supply chain using $LH_2$.

### 2.3.3. MCH Supply Chain

Figure 4 shows the supply chain for the use of MCH as an energy carrier, and Table 2 lists the specifications for each process ("Value" in Tables 2–5 means the specific values assumed for the parameters of each process in the supply chains). Parameters listed in Table 2 were set by collecting data from literature surveys [30,37,38].

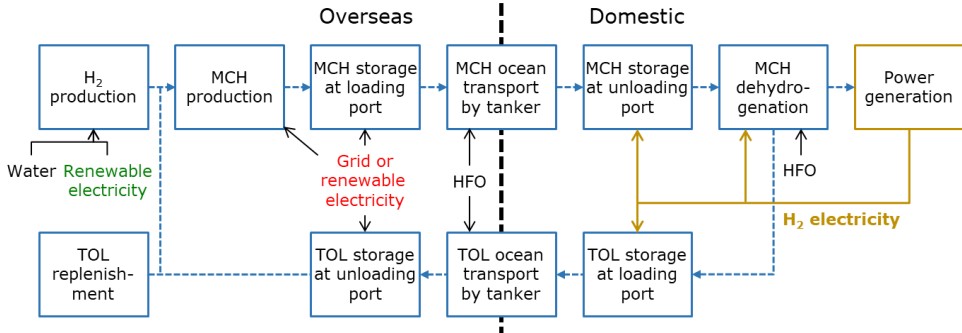

**Figure 4.** $H_2$ supply chain using MCH.

**Table 2.** Specifications of $H_2$ supply chain using MCH.

| Process | Parameter [Unit] | Value |
|---|---|---|
| MCH production | Reaction yield of $H_2$ addition to TOL [%] | 99.8 |
| | $H_2$ consumption rate [%] | 97.9 |
| MCH dehydrogenation | Conversion rate [%] | 95.0 |
| | Selectivity [%] | 99.9 |
| | $H_2$ yield [%] | 90.0 |
| TOL replenishment | Annual replenishment rate of TOL [%] | 3.0 |

**Table 3.** Specifications of $NH_3$ supply chain.

| Process | Parameter [Unit] | Value |
|---|---|---|
| $NH_3$ production from renewable hydrogen | Hydrogen loss rate during $NH_3$ synthesis [%] | 4.9 |
| | $NH_3$ loss rate during $NH_3$ liquefaction [%] | 0.1 |
| CO$_2$ capture | Rate of $CO_2$ capture [%] | 90 |
| | Monoethanolamine input [kg/t-$CO_2$ captured] | 2.41 |
| | Sodium hydroxide input [kg/t-$CO_2$ captured] | 0.13 |
| | Activated carbon input [kg/t-$CO_2$ captured] | 0.08 |
| $CO_2$ compression | $CO_2$ leakage from $CO_2$ compressor [t-$CO_2$/MW/year] | 23.2 |
| $CO_2$ transport | $CO_2$ leakage rate from pipelines [%] | 0.00367 |

**Table 4.** Specifications of coal supply chain.

| Process | Parameter [Unit] | Value |
|---|---|---|
| Coal mining and pretreatment | Electricity input [kWh/t-coal] | 13.2 |
| | Diesel input [liter /t-coal] | 3.88 |
| | Gasoline input [liter /t-coal] | 0.0309 |
| | Steel input [kg/t-coal] | 0.0170 |
| | Explosive input [kg/t-coal] | 2.81 |
| | Cement input [kg/t-coal] | 0.0108 |
| | Rubber input [kg/t-coal] | 0.000238 |
| | $CH_4$ emissions during coal mining [kg-$CO_2$/t-coal] | 121 |
| Coal inland transport | Diesel input [liter/t-coal] | 5.98 |
| Coal ocean transport | Heavy fuel oil (HFO) input [kg/t-coal] | 8.74 |

**Table 5.** Specifications of LNG supply chain.

| Process | Parameter [Unit] | Value |
|---|---|---|
| Natural gas mining | $CH_4$ emissions [kg-$CO_2$/t-natural gas] | 60.7 |
| LNG production | $CO_2$ emissions [kg-$CO_2$/t-natural gas] | 99.0 |
| | LNG input [kg/t-LNG] | 139 |
| LNG ocean transport | LNG input [kg/t-LNG] | 30.4 |
| | HFO input [kg/t-LNG] | 13.2 |

MCH is produced from hydrogen and toluene (TOL) and stored in cone-roof tanks at the loading port. TOL is produced from oil refining and naphtha reforming in Japan and is transported to a hydrogen-producing country by chemical tankers. The MCH produced using TOL is then transported from a hydrogen-producing country to Japan by chemical tankers. The petroleum product tanker transport of IDEA v2.2 was used to evaluate the $CO_2$ emission intensity of the chemical tankers. The imported MCH is stored in cone-roof tanks at the unloading port and then converted back to hydrogen and TOL by dehydrogenation. Hydrogen is supplied to the power plant, and the TOL is transported to the hydrogen-producing country for reuse. The heat required for dehydrogenation is assumed to be supplied by the combustion of HFO. TOL degraded by side reactions (formation of benzene and methane by demethylation reaction, paraffin by ring-opening reaction, cyclopentane by isomerization, and alkyl biphenyls by dimerization [28,37]) was assumed to be replenished. The probability distributions of electricity consumption due to MCH production, storage, dehydrogenation, and TOL storage were set as shown in Appendix A Table A2 based on published data [20,26].

*2.4. NH₃ Supply Chain*

Figure 5 shows the NH₃ supply chain, and Table 3 lists the specifications for each process. Parameters listed in Table 3 were set by collecting data from literature surveys [39–41].

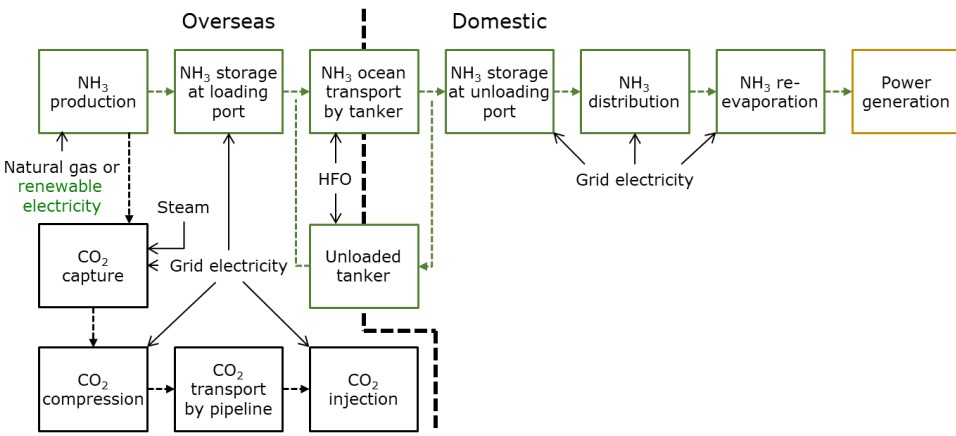

**Figure 5.** NH₃ supply chain.

### 2.4.1. NH₃ Production from Renewable Energy

NH₃ is synthesized and liquefied by the Haber–Bosch process from hydrogen produced by alkaline or solid polymer water electrolysis using solar PV power. Nitrogen was obtained via air separation. The consumption of electricity and pure water for hydrogen production was set as described in Section 2.3.1. The probability distributions of electricity consumption for nitrogen and NH₃ production were set as shown in Appendix A Table A3 based on published data [39,42–46].

### 2.4.2. NH₃ Production from Natural Gas

For the process of NH₃ production from natural gas, the input/output relationship was established based on the NH₃ production process of IDEA v2.2 from AIST, Tsukuba, Japan. In this process, natural gas is used as a feedstock for NH₃ and as a fuel for the reactor. The probability distributions of the utility natural gas and electricity consumption were set as shown in Appendix A Table A3.

Application of CCS is effective to mitigate $CO_2$ from NH₃ production process from natural gas. CCS consists of $CO_2$ capture, compression, pipeline transportation, and injection processes. Utility-derived $CO_2$ was recovered using the chemical absorption method (recovery rate: 90%). On the other hand, feedstock-derived $CO_2$ is emitted with high density from the NH₃ synthesis process; hence, a recovery process is not necessary. The recovered utility-derived $CO_2$ is compressed together with all the feedstock-derived $CO_2$, which is transported by pipeline to a $CO_2$ reservoir (transport distance: 50 km) and injected into the reservoir. The probability distributions of steam consumption due to $CO_2$ capture, electricity consumption due to $CO_2$ capture, compression, and injection were set as shown in Appendix A Table A3 based on published data [25,40,41,47–51]. As for the $CO_2$ emission intensity of steam required for chemical absorption, the probability distribution of Appendix A Table A3 was set with a minimum value of 0 (assuming steam can be supplied by waste heat) and the maximum value given by steam production from the HFO-fired boiler of IDEA v2.2 from AIST, Tsukuba, Japan.

### 2.4.3. NH₃ Storage, Transportation, and Regasification

The produced NH₃ is stored in tanks at the loading port and transported from the NH₃-producing country to Japan by NH₃ tankers (capacity: 38,000 m³ at a cruising speed of 17 knots). For the $CO_2$ emission intensity of the NH₃ tanker, the petroleum product tanker transportation of IDEA v2.2 was used by considering $CO_2$ emissions during round trips between Japan and the NH₃-producing country. After storage in tanks at the unloading port,

the imported $NH_3$ is transported by pipeline to the power plant, where it is re-gasified and used as fuel for power generation. The probability distribution of electricity consumption due to $NH_3$ storage and transportation was set as shown in Appendix A Table A3, based on published data [20].

### 2.5. Fossil Fuel Supply Chain

We calculated $CO_2$ emissions from the coal and LNG supply chains based on data from Ozawa et al. [20] and statistics on import destinations [52]. In this analysis, coal-fired and LNG-fired power plants are considered commercialized technologies, and the uncertainty of the fossil fuel supply chain is not considered.

Figure 6 shows the coal supply chain, and Table 4 lists the specifications for each process. Coal mined overseas first undergoes pre-processing. Methane emissions from coal mines and $CO_2$ emissions from consumables during mining (steel, explosives, etc.) were also considered. The coal is then transported by rail (fueled by diesel) to the port, where it is transported by coal carriers to power plants in Japan. For international transportation, $CO_2$ emissions for the round trip between Japan and the coal-producing country were considered, and the fuel was assumed to be HFO.

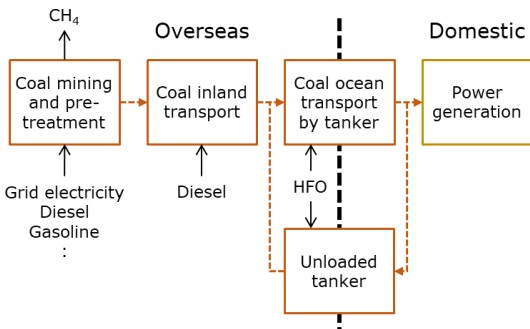

**Figure 6.** Coal supply chain.

Figure 7 shows the LNG supply chain, and Table 5 lists the specifications for each process. First, natural gas is extracted from overseas gas fields and liquefied. Consideration is also given to methane emissions from natural gas mining and $CO_2$ emissions from crude gas during liquefaction. LNG is then transported by a tanker to a power plant in Japan. For international transportation, $CO_2$ emissions for the round trip between Japan and the gas-producing country were considered, and the fuel was assumed to be boil-off gas and HFO.

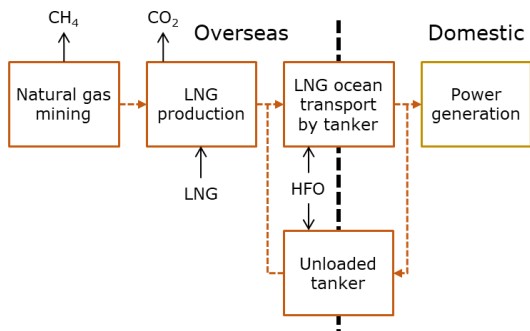

**Figure 7.** LNG supply chain.

### 2.6. Power Generation and CCS

Table 6 shows the specifications of the hydrogen and $NH_3$ power generation processes that were analyzed, and the thermal power generation process used for comparison. Five

types of power generation were examined: hydrogen–LNG co-firing, hydrogen mono-firing, $NH_3$–coal co-firing, $NH_3$–LNG co-firing, and $NH_3$ mono-firing. The co-firing ratio of hydrogen and $NH_3$ to existing fuels was set at 20% (based on a high heating value) in all cases. The efficiency of hydrogen mono-fired power generation was set to be lower than that of LNG-fired power generation [53]. The efficiencies of hydrogen co-firing power generation and $NH_3$ mono-firing/co-firing power generation were set to be the same as those of LNG-fired power generation. Thermal power generation was assumed to be ultra-supercritical coal-fired power generation (600 °C class) and LNG combined-cycle power generation (1500 °C class), and $CO_2$ emissions from coal ash disposal were also considered [23].

**Table 6.** Specifications of thermal power plants.

| Parameter [Unit] | $H_2$–LNG Co-Firing | $H_2$ Mono-Firing | $NH_3$–Coal Co-Firing | $NH_3$–LNG Co-Firing | $NH_3$ Mono-Firing | Coal | LNG |
|---|---|---|---|---|---|---|---|
| Co-firing ratio of $H_2$/$NH_3$ [%] (on HHV basis) | 20 | - | 20 | 20 | - | - | - |
| Capacity [MW] | 1000 | 1000 | 1000 | 1000 | 1000 | 1000 | 1000 |
| Capacity factor [%] | - | 70 | 70 | 70 | 70 | 70 | 70 |
| Efficiency (HHV) [%] | 53.0 | 51.3 | 39.6 | 53.0 | 53.0 | 39.6 | 53.0 |
| Auxiliary power ratio without CCS [%] | 2.08 | 2.08 | 5.06 | 2.08 | 2.08 | 5.06 | 2.08 |
| Electricity generated without CCS [MWh/year] | 6,004,454 | 6,004,454 | 5,821,721 | 6,004,454 | 6,004,454 | 5,821,721 | 6,004,454 |
| Auxiliary power ratio with CCS [%] | - | - | - | - | - | 36.89 | 15.03 |
| Electricity generated with CCS [MWh/year] | - | - | - | - | - | 3,869,743 | 5,210,228 |

HHV, higher heating value.

Table 7 lists the specifications for the CCS in thermal power plants ("Value" in Table 7 means the specific values assumed for the parameters of each process in the supply chains). Parameters listed in Table 7 were set by collecting data from literature surveys [40,41]. The emitted $CO_2$ is liquefied after being collected using the chemical absorption technique, transported by ship to a $CO_2$ reservoir (transport distance: 50 km), and injected into the reservoir. The electricity consumption required for $CO_2$ capture, liquefaction, and injection was assumed to be supplied by power generation by increasing the internal consumption rate.

**Table 7.** Specifications of carbon capture and storage (CCS) from thermal power plants [40,41].

| Process | Parameter [Unit] | Value |
|---|---|---|
| $CO_2$ capture | $CO_2$ capture rate [%] | 90 |
| | Steam input [GJ/t-$CO_2$ captured] | 3 |
| | Electricity input [kWh/t-$CO_2$ captured] | 23.6 |
| | Monoethanolamine input [kg/t-$CO_2$ captured] | 2.41 |
| | Sodium hydroxide input [kg/t-$CO_2$ captured] | 0.13 |
| | Activated carbon input [kg/t-$CO_2$ captured] | 0.08 |
| $CO_2$ compression | $CO_2$ leakage from $CO_2$ compressor [t-$CO_2$/MW/year] | 23.2 |
| $CO_2$ transport | HFO input for $CO_2$ transport by ship [kg/t-$CO_2$ transported] | 1.316 |
| $CO_2$ injection | HFO input for $CO_2$ injection [kg/t-$CO_2$ injected] | 3.666 |

*2.7. Uncertainty Analysis*

Uncertainty analysis aims at quantifying the variability of the calculation results attributed to the variation of the parameter input, and it helps robust decision making under uncertainty. As an example of uncertainty analysis, some financial studies focus on the effect of mitigating commodity pricing volatility in supply chains for supply chain risk management [54,55].

In terms of environmental profile of supply chains, there are two methods for dealing with uncertainty in the LCI analysis. One method is to conduct sensitivity analyses. The International Organization for Standardization (ISO) 14044 guidelines suggest that sensitivity analyses should include a wide range of factors to determine the influence of variations in assumptions, methods, and data [56]. An alternative option, the uncertainty analysis, employs probabilistic simulations based on the Monte Carlo method to evaluate the combined influence of multiple uncertain factors on the results. Here, probability distributions were assumed for the input parameters of the system. Repeated calculations with different input values yield a probability frequency distribution of the total GHG emissions from the entire system [57].

In this study, we performed a Monte Carlo simulation to calculate the life-cycle $CO_2$ emissions of hydrogen and $NH_3$ power generation. We treated energy consumption ($c_{i,j}$) and $CO_2$ emission intensity ($e_i$) for each process in the supply chains as uncertain parameters and calculated the life-cycle $CO_2$ emissions ($E$) stochastically using Equation (1). Based on the variations in the parameters collected by literature survey, probability distributions were assumed for each uncertain parameter, as shown in Appendix A Tables A1–A3. Either normal distribution or triangular distribution was applied for the parameters' probability density function. If a normal distribution was applied for a parameter $x$, the probability distribution of $f(x)$ was defined as Equation (2):

$$f(x) = \frac{1}{\sqrt{2\pi\sigma^2}} \exp\left\{-\frac{(x-\mu)^2}{2\sigma^2}\right\} \tag{2}$$

where $\mu$ and $\sigma$ were the mean and standard deviation of $x$, respectively.

If a triangular distribution was applied, $f(x)$ was defined as Equation (3):

$$\begin{aligned} f(x) &= \frac{2(x-\alpha)}{(\beta-\alpha)(\mu-\alpha)} \quad \text{for } \alpha \leq x < \mu \\ f(x) &= \frac{2}{(\beta-\alpha)} \quad \text{for } x = \mu \\ f(x) &= \frac{2(\beta-x)}{(\beta-\alpha)(\beta-\mu)} \quad \text{for } \mu < x \leq \beta \end{aligned} \tag{3}$$

where $\alpha$ and $\beta$ were the minimum and maximum values of $x$, respectively. In Appendix A, the probability distributions of the uncertain parameters assumed in this study are described in detail. Monte Carlo simulations were performed using Oracle's Crystal Ball, and the number of iterations was set to 100,000.

## 3. Results and Discussion

We conducted an LCI analysis of both hydrogen and $NH_3$ power generation (co-firing and mono-firing), while considering the uncertainties, to identify and understand the factors that are important for realizing low-carbon generation using hydrogen energy in the electric power sector. Hydrogen was assumed to be produced from renewable energy sources in AUS, NOR, and the UAE and stored and transported using $LH_2$ and MCH as carriers, while $NH_3$ was assumed to be produced from renewable energy or natural gas in the UAE. Parameters related to the supply chain were collected from the literature, and the Japanese life-cycle inventory database IDEA was used to calculate emissions. The probability distributions of each parameter were established using the energy consumption and $CO_2$ emissions of each process in the supply chain as variables, and the uncertainties of life-cycle $CO_2$ emissions were calculated stochastically using Monte Carlo simulation. We compared the life-cycle $CO_2$ emissions of hydrogen and $NH_3$ power generation obtained

from the Monte Carlo simulations with those of thermal power generation, as shown in Figures 8–10. The bars represent the mean of the results, and the error bars represent the 99.7% confidence interval (corresponding to 3σ if a normal distribution is assumed). Probability distributions of the life-cycle $CO_2$ emissions of hydrogen and $NH_3$ power generation are also illustrated in Appendix B Figures A1–A7 (see Appendix B).

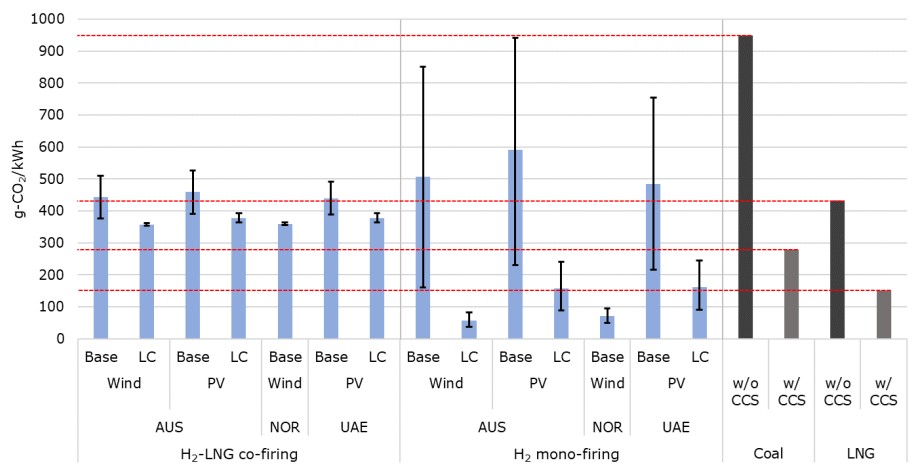

**Figure 8.** Life-cycle $CO_2$ emissions of hydrogen power generation using $LH_2$.

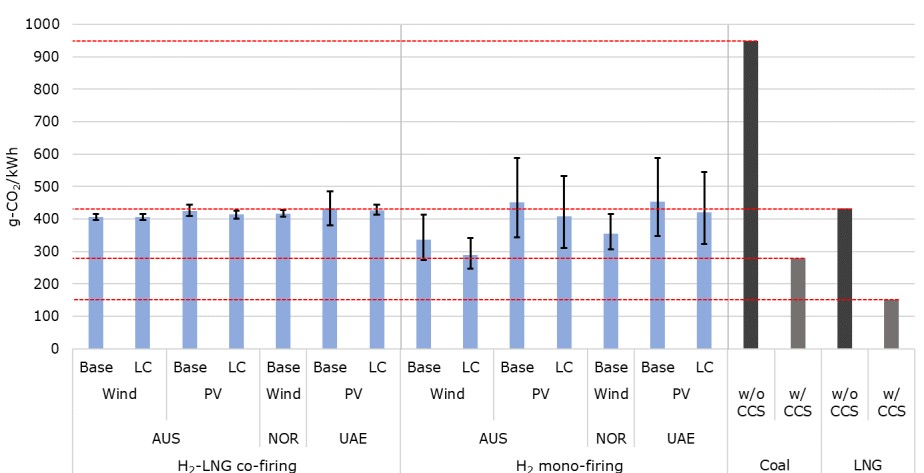

**Figure 9.** Life-cycle $CO_2$ emissions of $H_2$ power generation using MCH.

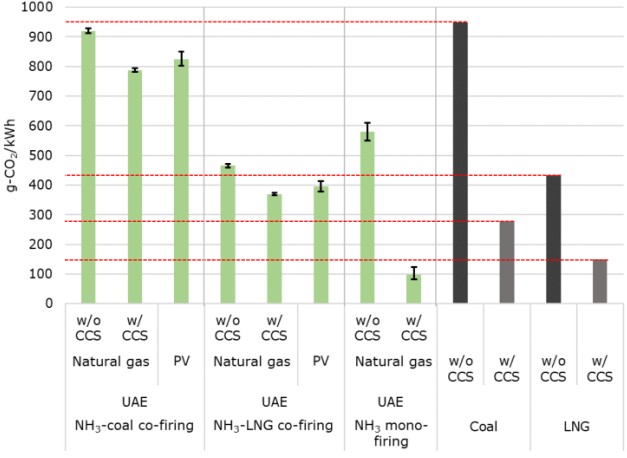

**Figure 10.** Life-cycle $CO_2$ emissions of $NH_3$ power generation.



### 3.1. Hydrogen Power Generation

Figure 8 shows life-cycle $CO_2$ emissions of hydrogen power generation when $LH_2$ is used as the energy carrier. The leftmost seven bars illustrate the emissions of hydrogen–LNG co-firing power generation (hydrogen co-firing ratio: 20%). If hydrogen is produced in AUS and the UAE and the base case is selected (grid electricity is used for overseas processes other than hydrogen production), the range of emission variation from hydrogen–LNG co-firing power generation includes those from LNG-fired power generation without CCS, which indicate that the two power generations are comparable with each other. On the other hand, in the low-carbon cases of AUS and the UAE (where all electricity consumption in overseas processes is covered by renewable energy) and in the case of hydrogen production in NOR, emissions are lower than those from LNG-fired power generation without CCS, indicating that the co-firing of hydrogen can reduce $CO_2$ emissions from LNG power plants. NOR has low emissions in the base case owing to the low $CO_2$ emission intensity of its grid electricity. It is also confirmed that emissions from hydrogen–LNG co-firing power generation is higher than those from coal- or LNG-fired power generation with CCS in all cases.

The differences between the cases become even more pronounced with hydrogen mono-firing. When hydrogen is produced in AUS and the UAE and the base case is selected, the range of emission variation is very large, and the emissions can reach those from coal-fired power generation in the worst case. The main reason for this is the effect of electricity consumption on the hydrogen liquefaction. As shown in Appendix A Table A2, the probability distribution of electricity consumption for hydrogen liquefaction has a large standard deviation. If grid electricity from AUS and the UAE (having high $CO_2$ emission intensities) is used for hydrogen liquefaction, the uncertainty attributed to this process will affect the emissions of the entire supply chain.

This uncertainty can be reduced by selecting a low-carbon case. In the low-carbon case of solar PV in AUS or the UAE, $CO_2$ emissions from electricity consumption are reduced by using renewable energy for hydrogen liquefaction, while life-cycle $CO_2$ emissions are lower than those from coal-fired power generation with CCS and comparable to LNG-fired power generation with CCS. If hydrogen is produced from wind power in AUS and the low-carbon case is chosen, or if hydrogen is produced in NOR, emissions will be lower than those of LNG-fired power generation with CCS.

Thus, if $LH_2$ is used as the energy carrier, the life-cycle $CO_2$ emissions of hydrogen power generation can be reduced by using renewable energy and grid electricity with low $CO_2$ emission intensity for hydrogen liquefaction.

Figure 9 shows life-cycle $CO_2$ emissions of hydrogen power generation when MCH is used as the energy carrier. In the case of hydrogen–LNG co-firing power generation (at a hydrogen mixed combustion ratio of 20%), the range of emission variation is comparable to those of LNG-fired power generation without CCS, except in cases where hydrogen is produced by wind power in AUS. Even in the AUS wind cases, the superiority to LNG-fired power generation is limited, indicating that the effect of $CO_2$ emission reduction of hydrogen co-firing is smaller when MCH is used as the energy carrier rather than $LH_2$. On the other hand, a comparison of hydrogen mono-firing with LNG-fired power generation without CCS shows that the emissions from hydrogen mono-firing are lower when hydrogen is produced by wind power in AUS and NOR. Especially if hydrogen is produced by wind power in AUS and the low-carbon case is selected, emissions is comparable to those from coal-fired power generation with CCS.

A comparison of the base case with the low-carbon case shows no significant difference in terms of emissions, which indicates that the effect of low-carbon electricity consumption in the overseas process is small because the power consumption is smaller than that of $LH_2$. In addition, a large amount of heat is required during MCH dehydrogenation, and $CO_2$ emissions from heat consumption account for 0~45% of the total. If the fuel for the dehydrogenation reaction is changed from HFO to city gas, $CO_2$ emissions from heat consumption can be reduced by ~20%. These results suggest the possibility of reducing

$CO_2$ emissions by using a low-carbon fuel or implementing waste-heat utilization in the MCH dehydrogenation process.

### 3.2. $NH_3$ Power Generation

Figure 10 shows life-cycle $CO_2$ emissions of $NH_3$ power generation. The result for $NH_3$–coal co-firing (at an $NH_3$ co-firing rate of 20%) shows that the range of emission variation is lower than those from coal-fired power generation without CCS in all cases, indicating that $NH_3$ co-firing is effective in reducing $CO_2$ emissions from coal power plants. In contrast, when comparing $NH_3$–LNG co-firing (at an $NH_3$ co-firing rate of 20%) and LNG-fired power generation without CCS, the emissions from $NH_3$–coal co-firing are lower if CCS is implemented, or when solar PV power is used in $NH_3$ production process. This result indicates that environmental benefits of $NH_3$ co-firing in LNG power plants can be gained when a low-carbon $NH_3$ production process is selected.

For $NH_3$ mono-firing, it is observed that the range of emission variation without CCS during $NH_3$ production from natural gas is higher than those of LNG-fired power generation without CCS, while the range of emission variation with CCS is lower than those of LNG-fired power generation with CCS. The latter result implies that the effect of $CO_2$ emission reduction of CCS during $NH_3$ production is greater than that of CCS during power generation, because the $NH_3$ production process emits highly pure $CO_2$ derived from the raw material and CCS is easier to implement than in thermal power generation. These results show that the use of CCS in the $NH_3$ production process can reduce the life-cycle $CO_2$ emissions of $NH_3$ power generation.

### 3.3. Summary of Results

Figure 11 and Table 8 summarize representative results on life-cycle $CO_2$ emissions of hydrogen and $NH_3$ power generation. The results show that life-cycle $CO_2$ emissions of hydrogen mono-firing power generation using $LH_2$ should be lower than those of LNG-fired power generation with CCS if hydrogen is produced from wind power in AUS and the low-carbon case is chosen, or if hydrogen is produced in NOR. Since electricity consumption for hydrogen liquefaction has a wide range of uncertainty, the total emissions can be decreased by utilizing low-carbon electricity for this process. Therefore, large-scale renewable power plants are required for supplying electricity to hydrogen production and liquefaction processes in hydrogen-producing countries. The results also indicate that life-cycle $CO_2$ emissions of $NH_3$ mono-firing power generation should be lower than those of LNG-fired power generation with CCS if CCS is available for $NH_3$ production process in the UAE. Since the $CO_2$ from $NH_3$ synthesis process has a high density, $CO_2$ capture from $NH_3$ production process is more efficient than post-combustion $CO_2$ capture from power plants.

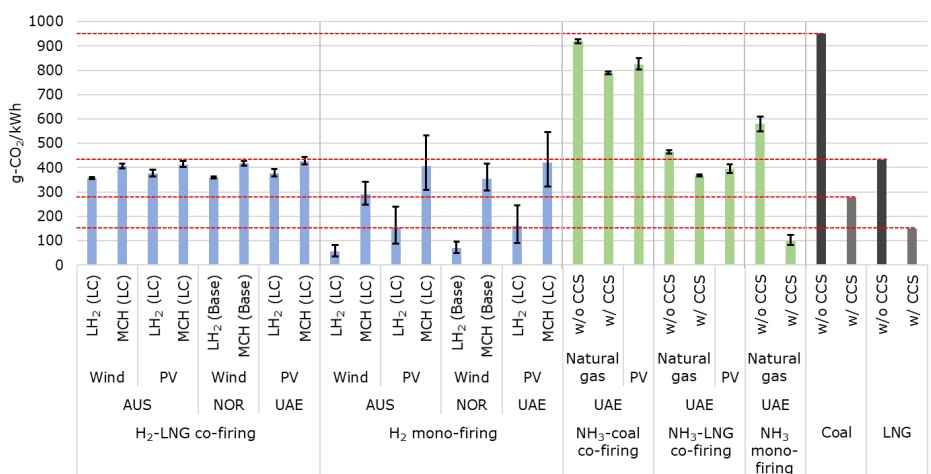

**Figure 11.** Representative results on life-cycle $CO_2$ emissions of $H_2$ and $NH_3$ power generation.

**Table 8.** Representative results on life-cycle $CO_2$ emissions of $H_2$ and $NH_3$ power generation.

| Power Generation | $H_2$/$NH_3$-Producing Country | Energy Source | Energy Carrier | Case | Life-Cycle $CO_2$ Emissions Mean (99.7% CI) [g-$CO_2$/kWh] |
|---|---|---|---|---|---|
| $H_2$–LNG co-firing | AUS | Wind | $LH_2$ | LC | 357.3 (353.4–362.2) |
| | | | MCH | LC | 405.9 (396.5–415.8) |
| | | PV | $LH_2$ | LC | 376.5 (363.5–392.5) |
| | | | MCH | LC | 413.4 (401.5–426.3) |
| | NOR | Wind | $LH_2$ | Base | 359.9 (355.9–364.7) |
| | | | MCH | Base | 416.6 (407.0–426.8) |
| | UAE | PV | $LH_2$ | LC | 377.2 (363.9–393.3) |
| | | | MCH | LC | 427.4 (414.1–443.4) |
| $H_2$ mono-firing | AUS | Wind | $LH_2$ | LC | 56.9 (36.2–82.3) |
| | | | MCH | LC | 289.7 (247.2–342.4) |
| | | PV | $LH_2$ | LC | 156.5 (89.1–240.3) |
| | | | MCH | LC | 407.9 (310.2–532.6) |
| | NOR | Wind | $LH_2$ | Base | 70.1 (49.4–95.2) |
| | | | MCH | Base | 355.3 (306.3–416.0) |
| | UAE | PV | $LH_2$ | LC | 160.7 (91.3–244.1) |
| | | | MCH | LC | 421.5 (322.4–545.2) |
| $NH_3$–coal co-firing | UAE | Natural gas | $NH_3$ | w/o CCS | 919.9 (911.6–928.3) |
| | | | | w/CCS | 786.7 (782.2–793.6) |
| | | PV | $NH_3$ | | 824.1 (801.8–849.5) |
| $NH_3$–LNG co-firing | UAE | Natural gas | $NH_3$ | w/o CCS | 465.3 (459.2–471.4) |
| | | | | w/CCS | 368.1 (364.8–373.2) |
| | | PV | $NH_3$ | | 395.4 (379.2–413.9) |
| $NH_3$ mono-firing | UAE | Natural gas | $NH_3$ | w/o CCS | 579.9 (550.0–610.5) |
| | | | | w/CCS | 98.7 (82.2–123.7) |
| Coal | | | | w/o CCS | 949.3 |
| | | | | w/CCS | 278.1 |
| LNG | | | | w/o CCS | 432.1 |
| | | | | w/CCS | 150.4 |

AUS, Australia; CI, confidence interval; NOR, Norway; LC, low-carbon; PV, photovoltaic; UAE, United Arab Emirates.

This study focuses on the environmental benefit of using hydrogen and $NH_3$ produced from renewable energy or natural gas as a power generation fuel. The LCI and its uncertainty analysis techniques used in this study can also be applied to other technology value chains where technological parameters are available. Moreover, the Monte Carlo method can be applied for uncertainty analysis of procurement cost for any technology value chain by assuming investment cost, operation and management cost, and commodity prices for each process.

## 4. Conclusions

This study evaluated uncertainties regarding the environmental benefits of hydrogen use in the electricity sector by applying the Monte Carlo method for LCI analysis. The results show that the life-cycle $CO_2$ emissions of hydrogen power generation using $LH_2$ is influenced by the uncertainty of electricity consumption during hydrogen liquefaction, whereas using low-carbon power sources (such as renewable energy for this process) can reduce uncertainty and $CO_2$ emissions. In the case of using MCH, while $CO_2$ emissions are not significantly affected by the power consumption of overseas processes, it is suggested that $CO_2$ emissions be reduced by implementing low-carbon fuel and waste-heat utilization during MCH dehydrogenation. The life-cycle $CO_2$ emissions of $NH_3$ power generation are significantly affected by a low-carbon $NH_3$ production process, and in particular, CCS during $NH_3$ production has a greater $CO_2$ emission reduction effect than CCS during power generation.

An assessment of the entire hydrogen energy supply chain provides important insights for realizing a significant reduction in the carbon emissions of hydrogen energy. In this study, we focused on the production of hydrogen and $NH_3$ using renewable energy or natural gas. In the future, it may be necessary to conduct similar analyses of lignite gasification and hydrogen production using domestic resources to examine various possibilities for realizing a low-carbon energy system through hydrogen energy. Similarly, we note

that conducting life-cycle analysis to include economic, social, and other environmental impacts as a part of multiple criteria analyses will also be increasingly important for the deliberation of hydrogen's potential role as an energy medium [58–61].

**Author Contributions:** Conceptualization, Y.K.; methodology, A.O., and Y.K.; software, A.O.; formal analysis, A.O.; investigation, A.O.; resources, A.O. and Y.K.; data curation, A.O. and Y.K.; writing—original draft preparation, A.O.; writing—review and editing, Y.K.; visualization, A.O.; supervision, Y.K.; project administration, Y.K.; funding acquisition, Y.K. All authors have read and agreed to the published version of the manuscript.

**Funding:** This research received no external funding.

**Institutional Review Board Statement:** Not applicable.

**Informed Consent Statement:** Not applicable.

**Data Availability Statement:** Not applicable.

**Conflicts of Interest:** The authors declare no conflict of interest.

**Abbreviations**

| | |
|---|---|
| AIST | National Institute of Advanced Industrial Science and Technology |
| AUS | Australia |
| CCS | Carbon capture and storage |
| CO2 | Carbon dioxide |
| GHG | Greenhouse gas |
| CH4 | Methane |
| CRIEPI | Central Research Institute of Electric Power Industry |
| HFO | Heavy fuel oil |
| HHV | Higher heating value |
| IDEA | Inventory Database for Environmental Analysis |
| IEA | International Energy Agency |
| IRENA | International Renewable Energy Agency |
| ISO | International Organization for Standardization |
| JHFC | Japan Hydrogen & Fuel Cell Demonstration |
| JPEC | Japan Petroleum Energy Center |
| JRC-IET | Joint Research Centre Institute for Energy and Transport |
| LC | Low-carbon |
| LCI | Life-cycle inventory |
| $LH_2$ | Liquid hydrogen |
| LNG | Liquefied natural gas |
| MCH | Methylcyclohexane |
| METI | Ministry of Economy, Trade and Industry |
| NETL | National Energy Technology Laboratory |
| $NH_3$ | Ammonia |
| NOR | Norway |
| $NO_x$ | Nitrogen oxide |
| PDF | Probability density function |
| PV | Photovoltaic |
| RD&D | Research, development, and demonstration |
| RITE | Research Institute of Innovative Technology for the Earth |
| SD | Standard deviation |
| TOL | Toluene |
| UAE | United Arab Emirates |

**Appendix A**

This section describes the probability distributions of the parameters related to the hydrogen supply chains. This study followed the method described in a previous study [19]. First, the variations in a parameter $X$ $x = \{x_1, \cdots, x_n\}$ are collected by liter-

ature surveys, and their mean ($\overline{x}$) and standard deviation ($s$) are obtained as shown in Equations (A1) tand (A2):

$$\overline{x} = \frac{1}{n}\sum_{i=1}^{n} x_i \tag{A1}$$

$$s = \sqrt{\frac{1}{n-1}\sum_{i=1}^{n}(x_i - \overline{x})^2} \tag{A2}$$

It is assumed that the parameter $X$ follows a normal distribution with mean $\overline{x}$ and standard deviation $s$, if $s$ is equal to or smaller than one-third of $\overline{x}$ ($s \leq \frac{\overline{x}}{3}$). On the other hand, a triangular distribution is assumed for the parameter $X$, and its median, and lower and upper limits are set as the mean, minimum, and maximum values of $x$, if $s$ is larger than one-third of $\overline{x}$ ($s > \frac{\overline{x}}{3}$), which is due to the prevention of energy input or $CO_2$ emission intensity from taking a negative value. If only one data point can be obtained for a parameter ($x = \{x_1\}$), it is assumed that the parameter follows a normal distribution with mean $x_1$ and standard deviation $\frac{x_1}{10}$. Tables A1–A3 show the probability distributions set in this study.

**Table A1.** Uncertainty parameters for renewable power generation.

| Parameter [Unit] | PDF | Mean | SD | Reference(s) |
|---|---|---|---|---|
| $CO_2$ emissions for wind power in AUS [g-$CO_2$/kWh] | Normal | 12.4 | 1.2 | [20,23,27] |
| $CO_2$ emissions for solar PV in AUS [g-$CO_2$/kWh] | Normal | 39.2 | 3.9 | [20,23,27] |
| $CO_2$ emissions for wind power in NOR [g-$CO_2$/kWh] | Normal | 12.1 | 1.2 | [20,23,27] |
| $CO_2$ emissions for solar PV in UAE [g-$CO_2$/kWh] | Normal | 39.9 | 4.0 | [20,23,27] |

AUS, Australia; NOR, Norway; PDF, probability density function; PV, photovoltaic; SD, standard deviation; UAE, United Arab Emirates.

**Table A2.** Uncertainty parameters for $H_2$ supply chain.

| Parameter [Unit] | PDF | Mean | SD | Min. | Max. | Reference(s) |
|---|---|---|---|---|---|---|
| Electricity input for hydrogen production via water electrolysis [kWh/Nm$^3$-H$_2$] | Normal | 4.88 | 0.80 | | | [28,29] |
| Electricity input for liquefaction of gaseous hydrogen to produce LH$_2$ [kWh/Nm$^3$-H$_2$] | Normal | 0.906 | 0.244 | | | [30–35] |
| Electricity input for LH$_2$ storage at loading port [kWh/Nm$^3$-H$_2$] | Normal | 0.055 | 0.006 | | | [18] |
| LH$_2$ Boil-off rate during ocean transport [%/day] | Triangular | 0.30 | | 0.20 | 0.40 | [30,31] |
| Electricity input for LH$_2$ storage at unloading port [kWh/Nm$^3$-H$_2$] | Normal | 0.017 | 0.002 | | | [18] |
| Electricity input for MCH production [kWh/t-MCH] | Triangular | 40.68 | | 7.54 | 93.30 | [30,31,36] |
| Electricity input for MCH storage at loading/unloading port [kWh/t-MCH] | Normal | 0.915 | 0.120 | | | [30,31] |
| Electricity input for MCH dehydrogenization [kWh/Nm$^3$-H$_2$] | Normal | 0.310 | 0.058 | | | [30,31,36] |
| Electricity input for TOL storage at loading/unloading port [kWh/t-TOL] | Normal | 0.915 | 0.120 | | | [30,31] |

LH$_2$, liquid hydrogen; MCH, methylcyclohexane; PDF, probability density function; PV, photovoltaic; SD, standard deviation; TOL, toluene.

**Table A3.** Uncertainty parameters for NH₃ supply chain.

| Parameter [Unit] | PDF | Mean | SD | Min. | Max. | Reference(s) |
|---|---|---|---|---|---|---|
| Electricity input for nitrogen separation from air [kWh/Nm³-N₂] | Triangular | 0.249 | | 0.195 | 0.380 | [43,44,46] |
| Electricity input for NH₃ production from renewable hydrogen [kWh/kg-NH₃] | Triangular | 0.457 | | 0.177 | 0.849 | [39,42,45,46] |
| Natural gas input as a fuel for NH₃ production from natural gas [MJ-NG(HHV)/kg-NH₃] | Normal | *a | *b | | | [25] |
| Electricity input for NH₃ production from natural gas [kWh/kg-NH₃] | Normal | *a | *b | | | [25] |
| Steam input for CO₂ capture [GJ/t-CO₂ captured] | Normal | 2.48 | 0.70 | | | [40,41,47–51] |
| CO₂ emissions for steam input for CO₂ capture [g-CO₂/kg-steam] | Triangular | *c | | 0 | *d | [25] |
| Electricity input for CO₂ capture [kWh/t-CO₂ captured] | Normal | 23.6 | 2.4 | | | [40,41] |
| Electricity input for CO₂ compression [kWh/t-CO₂ captured] | Normal | 127.8 | 12.8 | | | [40,41] |
| Electricity input for CO₂ injection [kWh/t-CO₂ captured] | Normal | 7.0 | 0.7 | | | [40,41] |
| Electricity input for NH₃ storage at loading port [kWh/t-NH₃] | Normal | 0.83 | 0.08 | | | [20] |
| Electricity input for NH₃ storage at unloading port [kWh/t-NH₃] | Normal | 2.77 | 0.28 | | | [20] |
| Electricity input for NH₃ distribution [kWh/t-NH₃] | Normal | 0.57 | 0.06 | | | [20] |
| Electricity input for NH₃ re-evaporation [kWh/t-NH₃] | Normal | 0.022 | 0.002 | | | [20] |

*a Data retrieved from the NH₃ production process of IDEA v2.2. *b 10% of the data retrieved from the NH₃ production process of IDEA v2.2. *c 50% of the data retrieved from steam production by the HFO-fired boiler of IDEA v2.2. *d Data retrieved from steam production by the HFO-fired boiler of IDEA v2.2. HFO, heavy fuel oil; HHV, higher heating value; NG, natural gas; PDF, probability density function; SD, standard deviation.

## Appendix B

Figures A1–A7 illustrate probability distributions of life-cycle CO₂ emissions of hydrogen and NH₃ power generation.

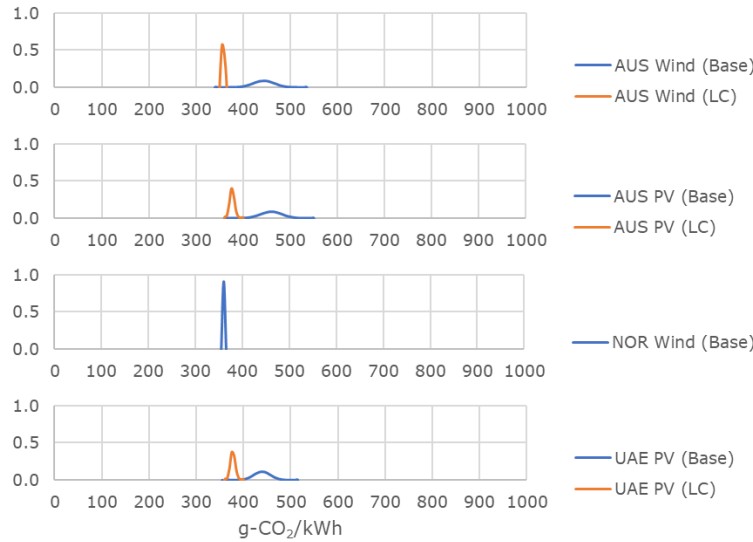

**Figure A1.** Probability distributions of life-cycle CO₂ emissions of hydrogen-LNG co-firing power generation when LH₂ is used as the energy carrier.

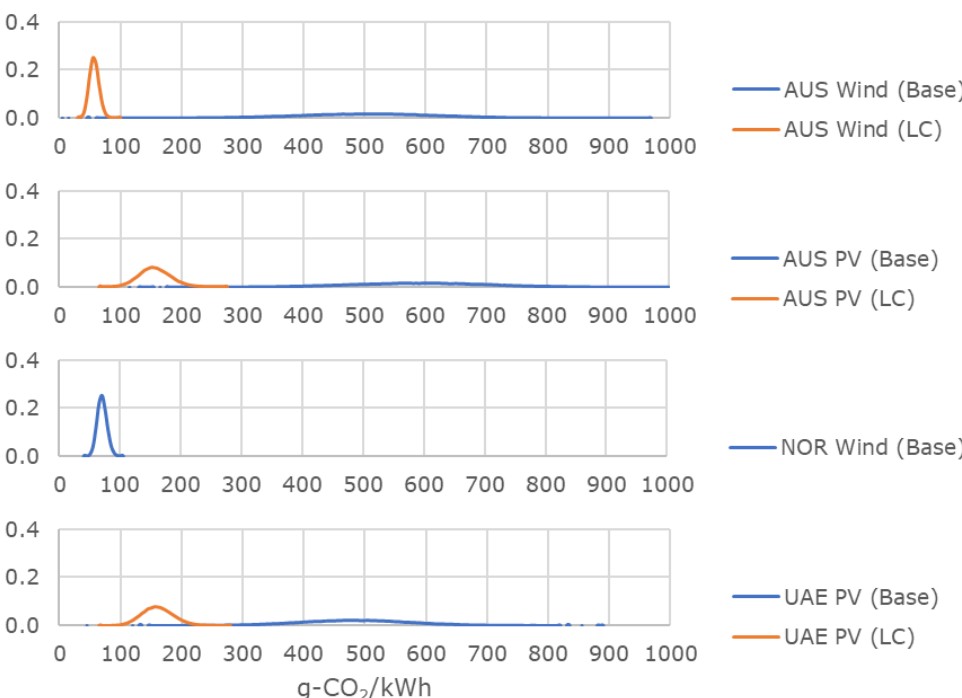

**Figure A2.** Probability distributions of life-cycle $CO_2$ emissions of hydrogen mono-firing power generation when $LH_2$ is used as the energy carrier.

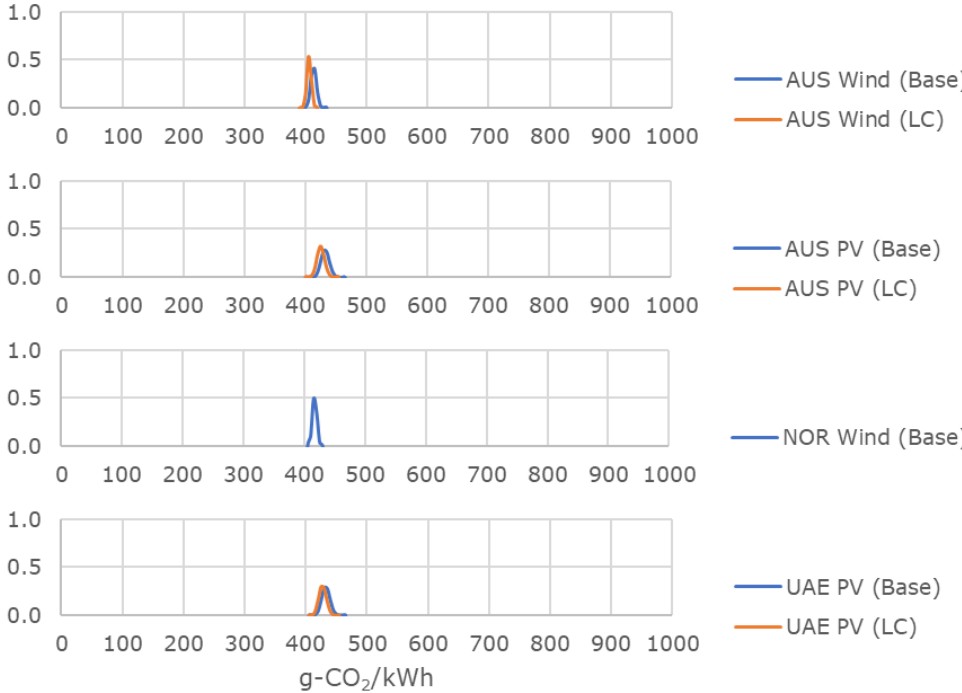

**Figure A3.** Probability distributions of life-cycle $CO_2$ emissions of hydrogen-LNG co-firing power generation when MCH is used as the energy carrier.

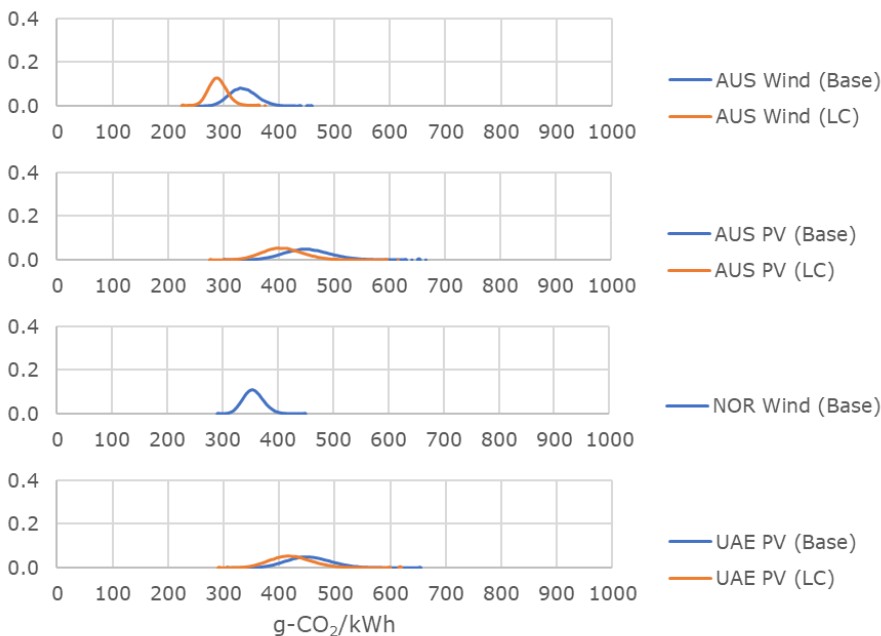

**Figure A4.** Probability distributions of life-cycle $CO_2$ emissions of hydrogen mono-firing power generation when MCH is used as the energy carrier.

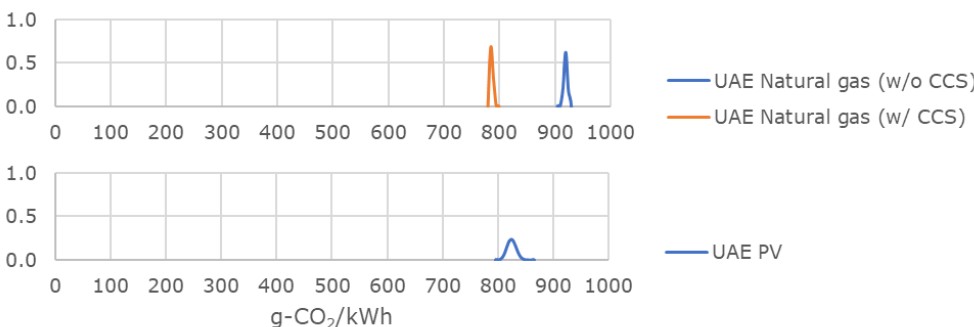

**Figure A5.** Probability distributions of life-cycle $CO_2$ emissions of $NH_3$-coal co-firing power generation.

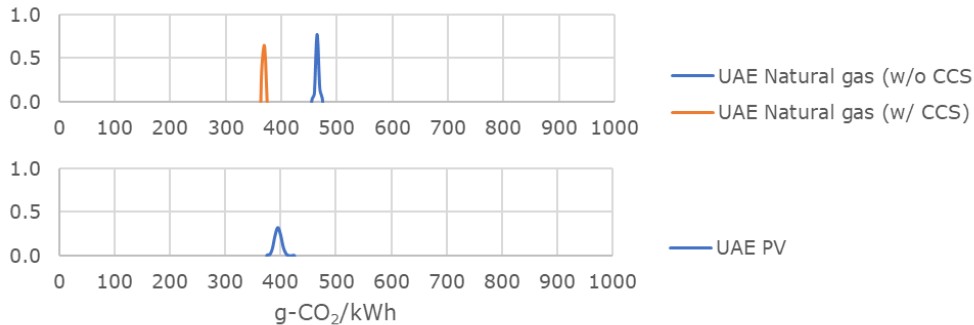

**Figure A6.** Probability distributions of life-cycle $CO_2$ emissions of $NH_3$-LNG co-firing power generation.

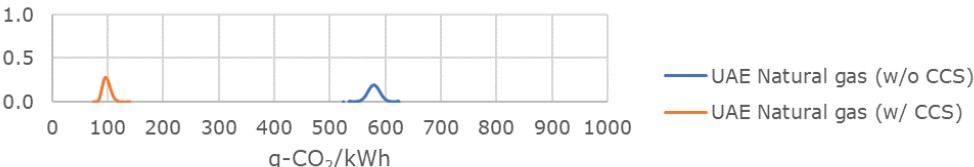

**Figure A7.** Probability distributions of life-cycle $CO_2$ emissions of $NH_3$ mono-firing power generation.

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
