# Peer review of "Assessing Uncertainties of Life-Cycle CO2 Emissions Using Hydrogen Energy for Power Generation"

_energies, doi:10.3390/en14216943_

Round 1
Reviewer 1 Report
The paper focuses on the evaluation of the complete environmental merits of hydrogen energy supply chain. The authors conducted life-cycle inventory analysis to calculate CO2 emissions attributed to the entire supply chain of hydrogen and NH3 power generation (co-firing and mono-firing) in Japan.
As for the methodology, I suggest clarifying and extend the part related to uncertainty analysis. I don’t see a definition of the uncertain variables and their modeling.
Also, I suggest adding some considerations about costs, with particular attention to uncertainty of price and volatility of commodity price (energy etc.) (Gaudenzi et al. 2020; Pellegrino et al., 2019; Steinberger-Wilckens and Sampson, 2019).
Suggested references
- Gaudenzi, B., Zsidisin, G. A., & Pellegrino, R. (2020). Measuring the financial effects of mitigating commodity price volatility in supply chains. Supply Chain Management: An International Journal.
- Pellegrino, R., Costantino, N., & Tauro, D. (2019). Supply Chain Finance: A supply chain-oriented perspective to mitigate commodity risk and pricing volatility. Journal of Purchasing and Supply Management, 25(2), 118-133.
- Steinberger-Wilckens, R., & Sampson, B. (2019). Market, Commercialization, and Deployment—Toward Appreciating Total Owner Cost of Hydrogen Energy Technologies. In Science and Engineering of Hydrogen-Based Energy Technologies(pp. 383-403). Academic Press.
Author Response
We appreciate your valuable comments. The manuscript has been modified according to your concerns. A point-by-point response to each comment is provided below.
Comment #1
As for the methodology, I suggest clarifying and extend the part related to uncertainty analysis. I don’t see a definition of the uncertain variables and their modeling.
Reply
We modified the subsection “2.7. Uncertainty analysis” (lines 337-367) to clarify what is done in this study. We added some description on the definition of uncertainty variables and their equation, as well as general explanation on uncertainty analysis for decision-making.
Comment #2
Also, I suggest adding some considerations about costs, with particular attention to uncertainty of price and volatility of commodity price (energy etc.) (Gaudenzi et al. 2020; Pellegrino et al., 2019; Steinberger-Wilckens and Sampson, 2019).
Suggested references
Gaudenzi, B., Zsidisin, G. A., & Pellegrino, R. (2020). Measuring the financial effects of mitigating commodity price volatility in supply chains. Supply Chain Management: An International Journal.
Pellegrino, R., Costantino, N., & Tauro, D. (2019). Supply Chain Finance: A supply chain-oriented perspective to mitigate commodity risk and pricing volatility. Journal of Purchasing and Supply Management, 25(2), 118-133.
Steinberger-Wilckens, R., & Sampson, B. (2019). Market, Commercialization, and Deployment—Toward Appreciating Total Owner Cost of Hydrogen Energy Technologies. In Science and Engineering of Hydrogen-Based Energy Technologies(pp. 383-403). Academic Press.
Reply
We added the references ([54],[55],[58]) accordingly.
Reviewer 2 Report
Dear authors, the manuscript is well-conceived and scientifically structured having some inadequacies. I shall highlight them and the authors might improve the quality and readability of this research paper accordingly.
The strengths of the manuscript: This research work deploys a vast repertoire of supply chain and technical analyses and related approaches of hydrogen energy and power generation to assess uncertainties of life-cycle CO2 emissions. This work details methodological rigour and controls while discreetly deploying the life-cycle inventory analysis. This research article reviews a wide array of literature and situates the prognosis related to the reduced uncertainty and CO2 emissions while using low-carbon power sources. The authors further mention the multiple factors responsible for the generation of low-carbon energy system through hydrogen energy and supply chain. The complex relationships among the diverse factors like NH3 mono-firing, NH3–LNG co-firing, NH3–Coal co-firing, the use of CCS and MCH and other associated parameters have been detailed in a rigorous manner.
- The methods are not correctly described and sufficiently informative to allow the replication of the research. The methodology of this research work should be described by a schematic and conceptual flowchart. I understand the inclusion of process flow charts showing stages within the system’s boundaries.
- The research questions appear late at the end of the introduction section. The authors may want to move this towards the beginning as it makes it easier for the readers to understand the context and the objective of the authors in explaining the background.
- The most relevant data-results should be summarized and demonstrated by a graph and a corresponding table.
- Please, highlight the outliers in the tables and graphs, where relevant.
- The paper does not compare to other methods. Please compare your approach with some commonly used case study approaches. I understand a combined approach, based on standard power generation technologies and supply chain analysis related to hydrogen economy, strategy and fuel cells.
- The techniques and/or models presented and mentioned in the manuscript require sufficient details (including calibration, sensitivity analysis and validation) to allow other researchers to develop and test the applications later on. Please include the parameters that I have mentioned here.
- The actual global contribution of this manuscript is fuzzy. The authors should clearly explain the intuitions behind their ideas, models and datasets. Additionally, more comparisons that show the advantages and the drawbacks of the proposed schema are needed.
- Please, briefly add future perspectives and further applied applications of this specific research work in the discussion section before the conclusions. It is not well interpreted to show how and why this work was undertaken relative to what is already known.
- The concepts, presented and mentioned in the manuscript must be supported by referential empirical models and associated details to allow other researchers to develop such frameworks later on.
- Please include the data and results of the Monte Carlo Simulation in a table and graph providing a brief excerpt on the theoretical premises.
Author Response
We appreciate your valuable comments. The manuscript has been modified according to your concerns. A point-by-point response to each comment is provided below.
Comment #1
The methods are not correctly described and sufficiently informative to allow the replication of the research. The methodology of this research work should be described by a schematic and conceptual flowchart. I understand the inclusion of process flow charts showing stages within the system’s boundaries.
Reply
We added Figure 1 and some descriptions into lines 119-132 to clarify the methodology of this study.
Comment #2
The research questions appear late at the end of the introduction section. The authors may want to move this towards the beginning as it makes it easier for the readers to understand the context and the objective of the authors in explaining the background.
Reply
To make clear the context and the objective of the article, we modified the abstract part (lines 12-14).
Comment #3
The most relevant data-results should be summarized and demonstrated by a graph and a corresponding table.
Reply
We added Figure 11 and Table 8 to summarize representative results on life-cycle CO2 emissions of hydrogen and NH3 power generation (lines 468-494).
Comment #4
Please, highlight the outliers in the tables and graphs, where relevant.
Reply
In our uncertainty analysis, input values shown in Tables A1-A3 don’t have any outliers, because these probability frequency distribution parameters are set using the collected data by literature surveys.
Comment #5
The paper does not compare to other methods. Please compare your approach with some commonly used case study approaches. I understand a combined approach, based on standard power generation technologies and supply chain analysis related to hydrogen economy, strategy and fuel cells.
Reply
We added some description of the difference between this study’s approach and other studies’ approach (lines 97-104). In most LCI studies, specific values are assumed for parameters regarding supply chains. However, most technologies that compose hydrogen supply chains for power generation are still in the development and demonstration stages, and their specification (e.g., energy input and CO2 emissions intensity) at the practical application stage have a certain degree of uncertainty, which have not been fully considered in previous research. This study aims to evaluate uncertainties regarding the environmental benefits of hydrogen use in the electricity sector by applying Monte Carlo method for LCI analysis.
Comment #6
The techniques and/or models presented and mentioned in the manuscript require sufficient details (including calibration, sensitivity analysis and validation) to allow other researchers to develop and test the applications later on. Please include the parameters that I have mentioned here.
Reply
We modified the subsection “2.7. Uncertainty analysis” (lines 337-367) to clarify what is done in this study. We added some description on the definition of uncertainty variables and their equation, as well as general explanation on uncertainty analysis for decision-making. We also described in detail the supply chain configuration assumed in this study.
Comment #7
The actual global contribution of this manuscript is fuzzy. The authors should clearly explain the intuitions behind their ideas, models and datasets. Additionally, more comparisons that show the advantages and the drawbacks of the proposed schema are needed.
Reply
Implication of this study toward lowering life-cycle CO2 emissions of hydrogen energy for power generation is described in the Conclusion part (lines 496-507). As written in lines 49-56, establishment of international hydrogen supply chains using liquid hydrogen energy carriers are essential for achieving hydrogen economy in Japan whose potential of producing hydrogen domestically is low. Most of the LCI analysis referred to in this article mainly focus on hydrogen that are produced domestically or regionally. Therefore, our study provide a benchmark for establishing international hydrogen supply chains toward decarbonizing energy systems of Japan.
Comment #8
Please, briefly add future perspectives and further applied applications of this specific research work in the discussion section before the conclusions. It is not well interpreted to show how and why this work was undertaken relative to what is already known.
Reply
We added some arguments on future perspectives and further applied application of this study into lines 480-486, 513-516.
Comment #9
The concepts, presented and mentioned in the manuscript must be supported by referential empirical models and associated details to allow other researchers to develop such frameworks later on.
Reply
We carefully revised the manuscript to clarify the concept of this study and added some detailed description on the methodology applied in this study (lines 337-367). We also explained modelling, equations and parameters assumed in this study (lines 119-132, 559-586).
Comment #10
Please include the data and results of the Monte Carlo Simulation in a table and graph providing a brief excerpt on the theoretical premises.
Reply
We added Figures B1-7 which illustrate the probability distributions of life-cycle CO2 emissions of hydrogen and NH3 power generation into Appendix B (lines 586-609).
Reviewer 3 Report
- This paper provides useful information for CCS technology. Therefore, I suggest that it can be published in Energies after minor revision.
- In Fig. 2, the author mentioned that H2 production depends on water and renewable electricity. Are these the popular technology in H2 production? The two methods need a considerable energy consumption. Other methods, such as hydrogen production from coal, are also a main H2 source. Therefore, more references for H2 supply chain should be provided in this study.
- Some references in this field should be cited (Fuel, 288, 119640; AIChE Journal, 67, pp. e17302; AIChE Journal, 66, pp. e16888).
- In Fig. 7, why the error bar exhibits a wide range in AUS.
- Please provide full name for all abbreviation.
- Except of energy consumption and CO2 emission, the lifetime or cost of various methods and instruments should be considered in this study.
- The “value” of all Tables is confusing. Can all values be compared between each other? Please provide a clear definition for it in this study.
- The authors should give an outlook on future research work.
I'm willing to review the revised manuscript.
Author Response
We appreciate your valuable comments. The manuscript has been modified according to your concerns. A point-by-point response to each comment is provided below.
Comment #1
In Fig. 2, the author mentioned that H2 production depends on water and renewable electricity. Are these the popular technology in H2 production? The two methods need a considerable energy consumption. Other methods, such as hydrogen production from coal, are also a main H2 source. Therefore, more references for H2 supply chain should be provided in this study.
Reply
Hydrogen that is produced from renewable energy (currently recognized as “green hydrogen” internationally) is considered to play a major role of low-carbon hydrogen in future. According to an IEA’s report, more than 300 Mt of green hydrogen will be produced by water electrolysis in 2050 to achieve global carbon-neutrality by 2050. We added explanation into lines 199-201 to clarify the focus of this study.
Comment #2
Some references in this field should be cited (Fuel, 288, 119640; AIChE Journal, 67, pp. e17302; AIChE Journal, 66, pp. e16888).
Reply
We added the references ([59]-[61]) accordingly.
Comment #3
In Fig. 7, why the error bar exhibits a wide range in AUS.
Reply
The reason is because of a wide range of uncertainty of electricity input for hydrogen liquefaction. As shown in Table A2, the probability distribution of electricity consumption for hydrogen liquefaction has a large standard deviation due to the uncertainty for liquefaction efficiency. Therefore, if grid electricity from AUS and the UAE (with high CO2 emission intensities) is used in this process, the emissions may differ widely, as noted in lines 407-411.
Comment #4
Please provide full name for all abbreviation.
Reply
List of Abbreviations are added into lines 525-557.
Comment #5
Except of energy consumption and CO2 emission, the lifetime or cost of various methods and instruments should be considered in this study.
Reply
In this study, we focus on the life-cycle CO2 emissions due to the operation of hydrogen supply chains and the emissions attributed to capital goods for the supply chains are out of scope, as stated in lines 112-114 and 122-124.
Comment #6
The “value” of all Tables is confusing. Can all values be compared between each other? Please provide a clear definition for it in this study.
Reply
We added the definition into lines 225-226. In this study, “value” means specific values assumed for the parameters of each process in the supply chains.
Comment #7
The authors should give an outlook on future research work.
Reply
We added some explanation on future research work into lines 511-516.
Round 2
Reviewer 1 Report
The authors addressed may previous comments
Reviewer 2 Report
Dear Authors, I am okay with the changes made. Thanks.